# Caesarean Section on Maternal Request-Ethical and Juridic Issues: A Narrative Review

**DOI:** 10.3390/medicina58091255

**Published:** 2022-09-10

**Authors:** Felice Sorrentino, Francesca Greco, Tea Palieri, Lorenzo Vasciaveo, Guglielmo Stabile, Stefania Carlucci, Antonio Simone Laganà, Luigi Nappi

**Affiliations:** 1Department of Medical and Surgical Sciences, Institute of Obstetrics and Gynecology, University of Foggia, 71121 Foggia, Italy; 2Department of Medicine, Surgery and Health Sciences, University of Trieste, 34100 Trieste, Italy; 3Unit of Gynecologic Oncology, ARNAS “Civico—Di Cristina—Benfratelli”, Department of Health Promotion, Mother and Child Care, Internal Medicine and Medical Specialties (PROMISE), University of Palermo, 90127 Palermo, Italy

**Keywords:** caesarean section, medico-legal, CDMR, tokophobia, defensive medicine

## Abstract

In recent decades, the rate of caesarean deliveries has increased worldwide. The reasons for this trend are still largely misunderstood and controversial among researchers. The decision often depends on the obstetrician, his beliefs and experience, the characteristics of the patients, the hospital environment and its internal protocols, the increasing use of induction of labor, the medico-legal implications, and, finally, the mother’s ability to request delivery by caesarean section without medical indication. This review aims to describe the reasons behind the increasing demand for caesarean sections by patients (CDMR) and strategies aimed at reducing caesarean section rates and educating women about the risks and benefits of CS.

## 1. Introduction

In recent decades, the rate of caesarean deliveries has increased worldwide. It is estimated to be 32% in the US, 32% in Oceania, 40% in Latin America, and 25% in Europe [1]. In Italy, the overall rate of caesarean deliveries is 35%, but in some regions it reaches 58–60% [2]. Today, caesarean section is one of the most common surgical procedures worldwide [3]. The reasons for this trend are still largely misunderstood and controversial among researchers. The decision often depends on the obstetrician, his beliefs and experience, the characteristics of the gravidity, the hospital environment and its internal protocols, the increasing use of induction of labor, the medico-legal implications and finally the mother’s ability to request delivery by caesarean section without medical indication. Cesarean birth on maternal request (CDMR) refers to a primary cesarean birth performed because the mother requests this method of delivery in the absence of a standard medical/obstetric indication for avoiding vaginal birth.

We must remember that medical indications for caesarean section include abnormal fetal presentations, maternal health problems, pathological cardiotocography registration, acute fetal distress, placental pathology etc. [4,5]. With the introduction of fetal cardiac monitoring, the interpretation of the fetal cardiac pattern became a central issue in obstetrics, and difficulties in interpretation were reflected in a higher rate of caesarean sections due to perceived “fetal distress” [6]. Another important issue in our generation is the fear of litigation [7]. In Spain, a retrospective study between 1986 and 2010 shows that most lawsuits related to obstetrics and gynecology concern obstetric issues (lawsuits about the mother’s health are lower than those about the baby’s health) [8]. Thus, clinical decisions about maternal and newborn health may be guided by fear of legal issues [9].

## 2. Materials and Methods

A search of the literature was conducted to identify the most relevant studies reported in English from 1984 until 2021 on PubMed MEDLINE electronic database. Based on the abstracts, we selected these studies, focusing on articles that concerned caesarean section on maternal request. The exclusion criteria were articles not in English and not relevant to the review and abstracts. The keywords used were “caesarean section”, “medico-legal”, “CDMR”, “tokophobia”, “defensive medicine”. Different combinations of the terms were used. Moreover, references in each article were searched to identify potentially missed studies.

## 3. Results and Discussion

### 3.1. Caesarean Section on Maternal Request (CDMR)-Ethical and Juridic Issues

Today, caesarean section seems to be a possible alternative to vaginal delivery for women. The questions are many. Is it ethical for a surgeon to perform an unnecessary procedure? The ACOG guidelines [10] recommend individualized care for women who wish to deliver by caesarean section and note that delivery by caesarean section is not recommended for women planning a multiple pregnancy because of the risks associated with multiple caesarean sections. Guidelines from Italy (Italian National Institute of Health-ISS), England (NICE), New Zealand, and Australia (RANZCOG) advocate caesarean section at the mother’s request, provided she is given appropriate information and advice [11,12,13]. Only the Canadian guidelines (SOCG) reject the option of CDMR and limit caesarean deliveries to standard indications [14] (Table 1).

However, the ethical question remains. If vaginal birth is the gold standard in the delivery room, why undergo surgery? In 2019, Romanis published a sincere defense of women’s rights in childbirth. She argues that CDMR in England is a “lottery” and women are often not given a choice by their doctors. She also points to women’s need to control their bodies during pregnancy and birth, which is seen as a unique experience that cannot simply be categorized as a ‘non-clinical need’. In fact, many women ask for a caesarean section because they are afraid of giving birth, because they have had a negative birth experience or because they have been sexually abused before. The doctor’s task is to show the patient some alternatives and support her in her decision. He should, according to the principle of harmlessness, advise against procedures that could be dangerous. Thus, a surgeon has the right to refuse an unnecessary operation [15]. The conflict between the principle of non-harmfulness and patient autonomy leads to legal problems. In Italy, a sample of gynecologists, midwives, lawyers and patients were recruited to ask five questions about caesarean section without medical indication. Patients perceived this decision as “therapeutic”, while gynecologists and midwives saw caesarean sections as a mistake and lawyers thought it was important to meet patients’ expectations. Patients also claim that the doctor can be legally prosecuted if a caesarean section was not performed, but they will not sue if there are complications in a CS [16]. Obstetric practice appears to be increasingly influenced by fear of litigation, with the appropriation of attitudes and procedures aimed at reducing discomfort [16]. In this atmosphere of mistrust, fear, and strife, it is not surprising that the doctor engages in behaviors aimed at reducing the risk of litigation. Thus, the rate of VBAC, operative deliveries, and breech deliveries decreased, and it is also believed that unnecessary caesarean sections could avoid the risks of labor [17]. The habit of “defensive medicine” leads to higher health care costs as unnecessary tests, blood tests, and surgeries are often requested, and patients are referred elsewhere. It has also been reported that some doctors avoid complicated patients or those they perceive as “argumentative” [18].

### 3.2. CDMR: Why?

Vaginal birth is a natural and important experience in a woman’s life. So, why do women request a caesarean section? The reasons seem to be multifactorial and include: anxiety and fear of the pain; prior poor labor experience; concern about fetal injury or death from vaginal birth; concerns about trauma to the pelvic floor; convenience of scheduled birth; positive attitudes toward cesarean in terms of convenience, short delivery time without labor pain. In details, one of the main reasons is the great fear of childbirth [19,20,21,22], which may be more pronounced in parous women than in nulliparous women [23]. Other risk factors include advanced maternal age, nulliparity and a high level of education [24,25,26] obesity, fear of pain and lack of confidence in vaginal birth. One reason may be fear of organ prolapse, sexual dysfunction and urinary incontinence [27]. A Brazilian study also found that care from the same doctor during pregnancy may be associated with CDMR [28]. In Italy, the “doctor factor” has also been pointed out, due to the Italian habit of seeing the same gynecologist throughout pregnancy, the high number of ultrasounds performed and the low habit of consulting midwives. This can lead to a strong medicalization of birth and pregnancy [29]. Some women are also concerned about the health of the baby and consider delivery by caesarean section “safer” for the newborn [30]. A large Brazilian multicenter study found that women with high education and economic means are more likely to deliver by CS, but middle-class women also want to deliver by caesarean section because they fear poor quality of care. Thus, a caesarean section is considered “best quality” compared to a vaginal delivery, which is perceived as negative or risky [31]. In some cases, infertility and a history of miscarriage can lead to this type of desire [32,33]. One of the most commonly cited variables in the literature is a previous negative birth experience, not only for instrumental deliveries, but in terms of an overall negative experience in the delivery room [34]. Some cultural beliefs, such as the belief that there is a lucky day for childbirth or a ‘golden hour’, may also influence the type of delivery” [35,36,37].

### 3.3. CDRM in the World

Globally, the rate of primary caesarean sections is increasing, but the rate of CDMR varies widely. One of the countries with the highest rate of CS is Brazil, with an average of 36% CS and women frequently planning a caesarean section even if they prefer a vaginal delivery, especially in private facilities [38], where the rate of surgical deliveries is about twice as high as in public facilities, where caesarean deliveries are even planned before admission [39,40]. As in Brazil, the rate of caesarean sections in Chile varies according to the type of insurance and the public or private hospital [41]. So, it is probably not the mother’s wish, but the relationship between public and private management of pregnancies that needs to be examined. Even in India, where the rate of caesarean sections is low, there are a high number of caesarean deliveries in private facilities. The reason for this is probably that doctors have to be present at all deliveries; so the need to schedule deliveries is a business model [42,43]. In China, there is a high rate of caesarean sections without medical indication, about 16% in secondary hospitals and 10% in tertiary care centers [44]. In Australia, the rate of CDRM is estimated to be about 17%. Moreover, 88% of medical specialists surveyed anonymously in this survey would perform CDMR, and 64.5% of residents also indicated that they would perform CDMR in their future career [45]. In the US, the CDMR rate is estimated to be around 3%, [46] although this is a low rate, there is no consensus on the diagnostic coding of CDMR [47]. A survey conducted in New York found that American women believe that “natural is better”, with the majority of women surveyed favoring vaginal delivery [48]. Between 7% and 8.2% of Swedish women wanted to deliver by caesarean section in 2002, and those who preferred surgical delivery belonged to a “vulnerable group”: anxious and depressed women with the usual risk factors (previous caesarean section, negative birth experiences) [49,50]. Moreover, 88% of the 413 Hungarian women preferred vaginal delivery to surgical delivery [51]. In Canada, too, few of the women surveyed would prefer a vaginal delivery, but the general belief was that a woman should be given the choice, regardless of the consequences in terms of complications or public health costs [52]. Only a small percentage of Thai women would prefer surgical delivery, believing that vaginal delivery guarantees a faster recovery [53] and a natural process. However, Thai women believe that even if the best choice is a vaginal delivery, the mother must be involved in the process of making decisions about the delivery route [54]. Another Thai survey found that only 68.9% of doctors would prefer vaginal delivery and more than 53% would perform CDMR [55]. Women in Singapore would also prefer a vaginal birth in principle, but in this survey, too, women had indicated their right to request a caesarean section [56]. The majority of Danish doctors would prefer a vaginal delivery in an uncomplicated pregnancy, but 40% of them believe that women have the right to choose the route of delivery themselves [57]. In Norway, in an interesting survey on the cost of CDMR, up to 40% of gynecologists said that women who want a caesarean section for non-medical reasons should contribute to the cost of the procedure [58]. It is clear from the global literature that while women do not prefer surgery, they demand their right to be involved in the decision about the route of delivery and that the doctor plays an important role, especially in a private setting. In 2015, the CDMR rate in Italy was estimated at 8.6% [34]. The reasons are the same as those described in the literature around the world: fear of delivery, previous negative experiences, less “traumatic” delivery for the child and convenience in scheduling the birth. 16.9% of southern Italian women would prefer to deliver by caesarean section [59]. A survey conducted among readers of a popular magazine found that up to 20% of women would prefer delivery by CS [60]. These surveys are in contrast to the results of the literature around the world. Does this mean that Italian women are more likely to deliver by caesarean section than in other parts of the world? In fact, in Italy, there are large differences between regions in terms of the percentage of caesarean sections. In 2012, for example, the C-section rate in Friuli-Venezia-Giulia was 21.4% and in Campania it was 62.1% [61]. There is also a big difference between public and private facilities: in Campania, the caesarean section rate in 2005 was an estimated 53.1% in private facilities, but 24.3% in public hospitals [62]. In a survey conducted in southern Italy (Sicily) in 2005, only 41% of doctors said they would perform a CDRM [63]. The reason for this preference for caesarean section is thus probably to be found in the complex socio-economic structure of Italian culture. On the one hand, there is the medico-legal aspect. Even though vaginal delivery is safer, doctors are often denounced for not performing CS [6]. Women retain the right to perform CS and in case of refusal they would sue for a possible complication during VB but not for a complication during a planned caesarean section. Italian doctors explain that they often practice “defensive medicine” and that the fear of litigation has changed their attitude, especially in facilities with a high proportion of caesarean sections [64]. One of the main reasons for preferring caesarean delivery over vaginal delivery is fear of childbirth, also known as tokophobia. The prevalence of this feeling varies greatly between countries [65,66] and is also difficult to categorize, ranging from moderate fear to extreme tokophobia. It is possible to identify some risk factors such as depression, anxiety, psychiatric disorders or sexual abuse, but usually this type of symptom is related to social background. Some women are simply afraid of the birth of their first child. They are afraid of labor and the possibility of losing control of their body, but even multiparous women may approach fear of childbirth having had an overall negative birth experience before. Although an obstetric complication in a previous delivery may lead to fear of childbirth, not all women who have experienced an obstetric complication view their birth experience as negative, indicating that perceptions are subjective [21].

### 3.4. What Is Tokophobia

A common feeling among pregnant women is fear of childbirth (FOC). Its prevalence is reported to be around 20–25% [67], while the prevalence of severe fear of childbirth (SFOC) is estimated at 14% [68]. SFOC also known as tokophobia [69] interferes with women’s daily routine and may affect their ability to actively participate in labor and birth [70]. Stressful and possibly traumatic experiences before pregnancy are associated with increased FOC during pregnancy. In particular, expectant mothers who have experienced emotional, psychological, or sexual abuse in childhood are more likely to experience this feeling, as are expectant mothers who have had a miscarriage prior to their current pregnancy [71]. Therefore, FOC may increase as the pregnancy progresses. The presence of gestational diabetes negatively affected women’s health experiences and resulted in increased FOC [72]. However, no association was found between other medical variables, such as type of conception (spontaneous or assisted conception), and FOC [73]. Finally, no study has compared the possible difference in FOC between mothers with singleton fetuses and mothers with multiple fetuses. The narrative review by Rondung et al. [74] reports that FOC is generally positively associated with anxiety (general, state, trait and sensitivity) or depression, but does not overlap. These findings suggest that FOC is a different syndrome related to the unknown, uncertain and uncontrollable [75]. The quality of the couple relationship has been shown to have an impact on FOC feelings during pregnancy. In particular, partner support is important, as low levels of couple satisfaction have been associated with an increased likelihood of FOC [76,77]. It is also important to distinguish between FOC and SFOC. Namely, while some level of FOC, if present, can be considered a physiological expression of the expectant mother’s feeling that something unknown and uncontrollable is happening [78], SFOC can instead be considered a clinical condition. SFOC has specific symptoms that can interfere with the normal functioning of the expectant mother and increase the risk to her health and that of her child. Higher levels of anxiety may be associated with a higher risk of dystocia, prolonged labor and emergency caesarean section [78,79]. FOC represents a specific dimension within a spectrum of pregnancy-related anxiety [80]. According to Bandura, a pregnant woman with a high level of FOC believes that she will not be able to cope successfully with childbirth (self-efficacy expectancy; SEE) and find appropriate strategies for the situation. If a woman with FOC is unable to mobilize her own resources, she cannot expect a favorable birth outcome (outcome efficacy expectancy-OEE). FOC can affect woman’s quality of life [81]. In cases where severe fear of conception occurs and leads to avoidance of tokos (Greek: birth), it is called primary tokophobia. Secondary tokophobia is defined as a phobic fear resulting from a stressful or even traumatizing experience at birth. While in physiological situations (i.e., low anxiety) relational variables (the quality of the couple relationship) may influence the woman’s feelings at the thought of childbirth, in the most extreme situations (i.e., high anxiety) anxiety is predicted exclusively by individual variables (depression). A woman with a negative mood or an emotional vulnerability has a higher probability to be depressed. Regarding socio-structural variables, there are contradictory results in previous studies suggesting that socio-demographic variables may not be predisposing factors for FOC per se, but may be associated with anxiety as moderators or mediators together with other, more psychological variables (e.g., anxiety, depression, couple relationship, etc.) or medical-obstetric variables related to the current pregnancy or the past (previous miscarriage, threatened miscarriage, gestational diabetes, etc.) [75]. Women with higher levels of FOC were significantly older and more likely to have reproductive risks and complications during their current pregnancy [82]. Low educational level, young age and unemployment were found to jointly predict FOC [79]. The severity of anxiety in pregnant women results from the interaction of all socio-demographic variables [75]. It is now recognized that childbirth can be traumatic and lead to symptoms of post-traumatic stress disorder (PTSD) in women [83]. Although rates vary by study, approximately 2–6% of women developed a PTSD profile directly related to their birth experience [84,85]. We should focus on the pre-natal period to avoid various postpartum disorders. Depressed pregnant women may develop intrusive thoughts of not being able to give birth and/or that their baby may die, leading to fear of childbirth and/or tokophobia [69]. Fear of childbirth is closely associated with the desire to have a caesarean section. In a Chinese study, 23.8% of women changed their preference from vaginal delivery to caesarean section after their first birth. Among these women, the rate of anxiety increased, especially in the case of emergency caesarean section, intrauterine growth restriction and anxiety [86,87]. However, a cohort study in Sweden showed that women who had a planned CS with a non-medical indication were overall less satisfied with their birth experience and decision-making process than women who had a planned vaginal birth [88]. The study also examined the level of depression in women requesting a caesarean section before and after birth. The level of depression did not change after birth, but women who requested a CS and then delivered vaginally had higher levels of post-traumatic stress disorder and postpartum depression [89,90]. Against this background, it seems clear that it is essential for the psychological and psychiatric integrity of women to take their wish for a caesarean section seriously. Women who planned to deliver by caesarean section and women who were hesitant about the mode of delivery reported significantly more FOC symptoms than women who planned to deliver vaginally. Since planning a caesarean section was a significant independent predictor of FOC, special attention should be paid to the results regarding caesarean section. We can speculate that a possible medical caesarean section is not a personal preference and could lead to an increase in FOC symptoms [91]. The treatment of fear of childbirth is still controversial. Certainly, talking to experienced doctors and midwives or having a therapeutic talk during pregnancy can be an option, as can birth preparation courses for couples [92]. A careful counseling may lead to a more positive birth experience and thus a lower incidence of anxiety in subsequent pregnancies [86]. The aim of doctors should probably not be to dissuade these patients from having a caesarean delivery, but to counsel and heal them appropriately so that they have a positive birth experience. Certainly, one strategy may be to inform women broadly about the possibility of early analgesia, to establish a one-to-one relationship with the midwife, and to consider the option of caesarean section if they do not respond to information, counseling, and psychotherapy.

### 3.5. Strategies to Avoid Cesarean Sections

Despite methodological limitations, individual or group psychoeducational sessions for nulliparous women or therapeutic discussions during pregnancy (in group or individual sessions) have the potential to strengthen women’s self-efficacy and reduce the number of caesarean sections due to FOC. The theoretical validation of an intervention deepens the understanding of psychological processes in women coping with severe FOC [93].

#### 3.5.1. Quality of Evidence High

Interventions for health professionals: use of clinical guidelines in conjunction with mandatory second opinion for indication of caesarean section; use of clinical guidelines in conjunction with audits and feedback on caesarean section practice; training of health professionals by opinion leaders (obstetricians/gynecologists).

#### 3.5.2. Quality of Evidence Low

Interventions targeting women or families: antenatal classes for mothers and couples, nurse-led relaxation training programs, psychosocial prevention programs for couples and psychoeducation.

Interventions targeting health care organizations or facilities: cooperative midwifery and obstetric model (where the obstetrician delivers at home 24 h a day without taking on competing clinical tasks) compared to a private model of care [92]. Evidence on the effectiveness of interventions directly targeting pregnant women is limited. There is some evidence that antenatal group therapy and antenatal classes can be effective in reducing caesarean section rates in low-risk pregnancies. However, the existing evidence is limited to small studies conducted in low-and middle-income countries (LMICs) where caesarean section rates were already high. The transferability of these findings to other settings and populations should be treated with caution. The introduction of clinical practice guidelines that require mandatory second opinions or are supported by local opinion leaders, as well as internal peer reviews provided to individual departments, may lead to a reduction in caesarean section rates. However, the evidence is weak and the costs and potential benefits of introducing these measures should be weighed [94].

### 3.6. Legal Point of View

The decision to use a CS or decision to delivery interval (DDI) is perhaps one of the most misunderstood and difficult issues facing obstetric teams today, and certainly one that arises in many cases of medical malpractice in obstetrics [95]. In recent years, litigation has become a major problem for OB/GYNs worldwide [7]. A history of litigation in obstetrics contributes to more defensive medicine and a higher likelihood of recommending a caesarean section, regardless of caps on non-economic damages. Clinicians who frequently worry about being sued were more likely to recommend caesarean section than those who “rarely worry about litigation” [95]. There has been a remarkable increase in medical malpractice lawsuits in Turkey and worldwide against OB /GYN. Turkey has serious problems with the high caesarean section rate, which was linked to medico-legal problems in a previous study [96]. A study also shows that defensive caesarean section is a common practice among obstetricians in Romania. Defensive medicine, in a general sense, is a term used to describe the actions taken by health professionals to reduce the likelihood of being sued rather than to help the patient. [97,98]. Defensive medicine not only harms the potential to treat patients, but also poses health risks. Defensive medicine disrupts the patient–doctor relationship and increases healthcare costs [99]. In the third millennium, there is a new “trend” in births: caesarean section (CS). Why the increasing rate? Firstly because of the later delivery date, secondly because of the safer anesthesia, and thirdly because of medical litigation. In Italy, about 38% of women are delivered by CS, with the rate being highest in the south of Italy (about 60% in Campania). The WHO stated in 1980 that 10% of CS was the gold standard, but now this rate is too low and cannot be achieved in most countries. Another aspect is that in the past the woman did not have many choices and the “medicine prescribed by the doctor” was the one the patient received. Today, however, with informed consent and greater participation by the woman, “patient’s medicine” is becoming more prevalent in medical practice. We demand that the gynecologist must be independent and have the power of decision. The gynecologist must not become a “victim” of his patients’ decisions, which may or may not be wrong. Therefore, the gynecologist should calmly point out to the patient what he or she believes is best and safest, based on his or her own experience, supported by the medical literature and medical guidelines [6]. The remaining question to be answered concerns what to do in cases where adequate information and persuasion fail, and a woman insists on a caesarean section contrary to the gynecologist’s recommendation. Many cases seem to be due to particular personal circumstances, such as psychosocial difficulties, previous negative experiences, and particular fears or anxieties about vaginal birth. We believe that in selected cases it is more beneficial to consider such personal circumstances and consent to caesarean delivery than to subject the woman to vaginal birth against her will. However, compliance with CDMR should remain the exception justified by special individual circumstances such as those mentioned above. Discussion of the mode of delivery should begin in early pregnancy to allow sufficient time for listening and counselling, seeking a second opinion, and if disagreement persists, timely referral to a colleague without compromising the patient’s care [100]. When a patient expresses a desire for an elective caesarean section, the obstetrician, in his capacity as the patient’s advocate, must help guide the patient through the maze of detailed medical information to a decision that respects both the patient’s autonomy and the physician’s obligation to optimize the health of both mother and newborn [101].

## 4. Conclusions

Nowadays, caesarean section is one of the most common surgical interventions. There are many reasons for this, including fear of medical malpractice lawsuits and the medicalization of birth and delivery. A lawsuit is a common episode in obstetric practice, but also a stressful event in medical career. It is therefore not surprising that most doctors try to avoid litigation. It is interesting to note that some women actually prefer to have a surgical delivery rather than a natural one. CDMR stands for caesarean section at the mother’s request, without medical or obstetric indication. The incidence of CDMR varies widely from country to country and region to region, ranging from 0.2 to 42% [34,46,102]. Although vaginal birth or ‘natural’ delivery is considered the standard of care for childbirth, in some cases caesarean section is considered the best quality service and is therefore desirable, even if there is no medical indication. In other cases, negative experiences lead women to opt for a surgical delivery because they fear that the fetus will be harmed or because they want a caesarean section for cultural reasons or fear of the unknown. Anxiety, stress, depression, and fear can seriously compromise a woman’s psychological integrity at such a delicate moment as childbirth. On the part of women, attention is paid to the conflict between the inevitable right of women to self-determination and the need to respect the independence of medical action, which must, however, always be consistent with the principle of beneficence. It should be emphasized that any acceptance of the maternal request for a caesarean section requires a specific informed consent, which clearly highlights the absence of a strictly clinical indication and information about this absence. It must also explicitly refer to the complication rates of caesarean section. In summary, the aim of healthcare personnel (both doctors and midwives) should be to provide appropriate advice to women, not with the aim of dissuading them from having a caesarean section, but, more importantly, to encourage the patient to make an informed decision and have an overall positive birth experience. Longitudinal studies should be developed to design strategies aimed at reducing the caesarean section rate and educating women about the risks and benefits of CS in their situation.

## Figures and Tables

**Table 1 medicina-58-01255-t001:** International guidelines and CDMR.

Guidelines		Position about CDMR
**ACOG**	** *2007* **	In favor, after appropriate information and counseling
**SOGC**	** *2009* **	Opposed
**NICE**	** *2011* **	In favor, after appropriate information and counseling
**RANZCOG**	** *2013* **	In favor, after appropriate information and counseling
**ISS**	** *2016* **	In favor, after appropriate information and counseling

CDMR: Cesarean birth on maternal request.

## Data Availability

No new data were created or analyzed in this study. Data sharing is not applicable to this article.

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
