# Peer review of "Caesarean Section on Maternal Request-Ethical and Juridic Issues: A Narrative Review"

_medicina, 2022, doi:10.3390/medicina58091255_

Round 1

Reviewer 1 Report

Review of article titled: “"Caesarean section on maternal request-ethical and juridic issues: a narrative review" by: "Felice Sorrentino, Francesca Greco, Tea Palieri, Lorenzo Vasciaveo, Guglielmo Stabile, Stefania Carlucci, Antonio Simone Laganà, Luigi Nappi"

The increasing rate of Cesarean section (C/S) has become a major issue for most health systems. In that context, a review of the ethical and juridic issues of cesarean section on maternal request is very important since more and more women “demand” to be delivered by C/S.

The article under review is a well written one but there are some methodological points that need to be addressed. Even though this is a narrative review, there are still some steps that need to be followed.

1.      What is the purpose of that review?

2.      Where were the articles retrieved from (electronic databases, journals, personal communication, etc.) and what was the time frame that was set for article retrieval? What keywords were they used?

3.      What were the criteria for the selection of the articles that were eventually used (inclusion-exclusion criteria)?

As for the setup of text itself, there are a few points that I would like to make:

1.      Sections 4 & 5 should probably be moved forward so as to show the burden of the phenomenon both worldwide but also in Italy.

2.      Section 3 could follow but I think that the possible answers to the question “why” should be stated more clearly.

3.      “Tokophobia” as a phenomenon is discussed both in section 5 (lines 178-207) but also in a separate section (section 6): I think that all that information should be condensed in one section.

4.      I think that a table presenting key points of the relevant guidelines of scientific societies would be useful.

5.      I believe that a “Conclusions” section is necessary, summarizing all the bulk of information: epidemiology, reasons, legal and ethical implications.

Reviewer 2 Report

This narrative review aims to summarize the reasons for the increasing request for cesarean section on mothers request. 

The topic is wide but has been adequately divided into paragraphs and well summarized.

Uniformize CSMR and CDMR in the manuscript.

Round 2

Reviewer 1 Report

The changes that were made to the structure of the article made it easier to read and understand. The addition of the "Conclusions" paragraph also contributed to that.